# Temporally Sparse Attack for Fooling Large Language Models in Time Series Forecasting

**Fuqiang Liu**[*] **& Sicong Jiang**
Department of Civil Engineering
McGill University
845 Sherbrooke St W, Montreal, Qc, H3A 0G4, Canada

## Abstract

Large Language Models (LLMs) have shown great potential in time series forecasting by capturing complex temporal patterns. Recent research reveals that LLM-based forecasters are highly sensitive to small input perturbations. However, existing attack methods often require modifying the entire time series, which is impractical in real-world scenarios. To address this, we propose a Temporally Sparse Attack (TSA) for LLM-based time series forecasting. By modeling the attack process as a Cardinality-Constrained Optimization Problem (CCOP), we develop a Subspace Pursuit (SP)–based method that restricts perturbations to a limited number of time steps, enabling efficient attacks. Experiments on advanced LLM-based time series models, including LLMTime (GPT-3.5, GPT-4, LLaMa, and Mistral), TimeGPT, and TimeLLM, show that modifying just 10% of the input can significantly degrade forecasting performance across diverse datasets. This finding reveals a critical vulnerability in current LLM-based forecasters to low-dimensional adversarial attacks. Furthermore, our study underscores the practical application of CCOP and SP techniques in trustworthy AI, demonstrating their effectiveness in generating sparse, high-impact attacks and providing valuable insights into improving the robustness of AI systems.

## 1 Introduction

Time series forecasting is a critical tool across various domains, including finance, traffic, energy management, and climate science. Accurate predictions of temporal patterns enable stakeholders to make informed decisions, optimize resources, and mitigate risks, thus playing a pivotal role in modern decision-making (Lim & Zohren, 2021; Liu et al., 2022b). By analyzing historical data to uncover trends, time series forecasting helps anticipate future events and take proactive actions.

Recently, Large Language Models (LLMs), originally designed for Natural Language Processing (NLP), have shown significant promise in capturing complex temporal dependencies across diverse scenarios (Garza & Mergenthaler-Canseco, 2023; Jin et al., 2024; Gruver et al., 2024). LLMs offer advanced capabilities, such as zero-shot forecasting, that allow them to generalize across various tasks without extensive retraining (Rasul et al., 2023; Ye et al., 2024; Liang et al., 2024). This positions LLMs as strong candidates for foundational models in time series forecasting. Pre-trained on vast and diverse datasets, these models leverage attention mechanisms to capture intricate temporal patterns and perform well on complex forecasting tasks (Devlin et al., 2019; Brown, 2020; Touvron et al., 2023; Liu et al., 2024a).

Despite these strengths, LLMs are known to be susceptible to adversarial attacks, raising concerns about their reliability in critical applications (Zou et al., 2023; Liu et al., 2024c). Adversarial attacks introduce subtle perturbations to input data, which can significantly degrade model performance. While LLM-based forecasters have demonstrated impressive accuracy in various tasks (Jiang et al., 2024), it remains uncertain whether decision-making processes can depend on these predictions in adversarial scenarios. Investigating the robustness of LLM-based models is therefore essential for ensuring their trustworthiness in real-world applications.

---

[*]Corresponding Author.

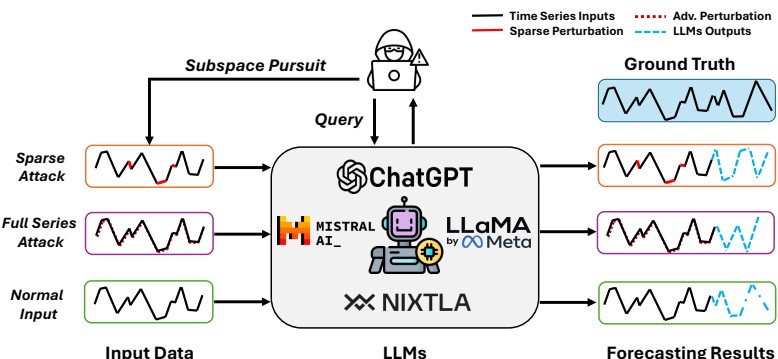

Figure 1: Temporally sparse black-box attack against LLMs in time series forecasting.

While adversarial attacks on machine learning models have been widely studied in image and NLP domains (Wei et al., 2018; Xu et al., 2020; Morris et al., 2020), attacking LLMs in time series forecasting presents unique challenges. First, ground truth values (i.e., future time steps) cannot be used in attacks to prevent information leakage. Second, accessing the internal parameters and structure of LLMs is often infeasible to attackers, requiring attacks to operate under strict black-box conditions. Recent studies have proposed targeted gradient-free optimization-based attacks to address these challenges (Liu et al., 2024b), but these methods remain impractical as they rely on perturbing the entire input time series. Consequently, this raises a critical question: **Is it possible to disrupt LLM-based forecasters by modifying only a small portion of the input time series?**

As shown in Figure 1, we address this question by developing a Temporally Sparse Attack (TSA) strategy tailored for highly constrained scenarios, where only a small subset of the input time series can be modified. We model the attack process as a Cardinality-Constrained Optimization Problem (CCOP) (Bhattacharya, 2009; Ruiz-Torrubiano et al., 2010), which applies sparse perturbations to selected time steps. To solve this CCOP, we propose a Subspace Pursuit (SP)-based method that leverages black-box query access to the target forecasting model. The TSA approach generates effective perturbations without requiring access to future data or internal model parameters, making it both practical and adaptable to real-world constraints.

Our evaluation covers three key types of LLM-based time series forecasting models, including six sub-models tested on four diverse real-world datasets. The results show that temporally sparse perturbations—affecting only 10% of the input data—can cause significant prediction errors, revealing a critical vulnerability in LLM-based forecasters. Even filter-based defense mechanisms struggle to mitigate these attacks due to their sparse and targeted nature. These findings underscore the need for more robust forecasting models that can resist adversarial manipulations and maintain reliability in real-world applications.

In conclusion, this study reveals the vulnerabilities of LLMs in time series forecasting under highly constrained conditions. The findings underscore the urgent need to address these vulnerabilities to develop LLMs that are not only accurate but also robust, thereby improving their practical applicability in high-stakes environments. Moreover, this work introduces CCOP and SP techniques into adversarial study, offering a novel and effective framework for modeling attack processes and generating temporally sparse perturbations. These contributions pave the way for future advancements in the robustness and reliability of LLM-based forecasting.

## 2 RELATED WORK

### 2.1 ATTACK ON LLMS

Adversarial attacks on LLMs have garnered significant attention, revealing how minor input manipulations can lead to substantial output alterations. These attacks are generally categorized into methods such as jailbreak prompting, where crafted prompts bypass safety guardrails to elicit unintended or

harmful responses (Wei et al., 2024); prompt injection, embedding adversarial instructions within benign prompts to manipulate outputs (Greshake et al., 2023; Xue et al., 2024; Shen et al., 2024); gradient-based attacks, which exploit internal model parameters to create minimally invasive input perturbations (Zou et al., 2023; Jia et al., 2024); and embedding perturbations, which subtly alter input embeddings to disrupt the model's internal representations (Schwinn et al., 2024).

While much of this research has focused on text-based tasks, the robustness of LLMs in non-textual domains like time series forecasting remains underexplored. Unlike static text, time series data is dynamic and continuously evolving, requiring perturbations that maintain the natural flow and coherence of the sequence. This dynamic nature introduces unique challenges for adversarial attacks, as traditional techniques designed for static inputs may not directly apply to temporal and sequential data. For instance, in static applications, true labels are readily available and play a crucial role in adversarial attack generation; however, in forecasting applications, obtaining future true labels is infeasible.

## 2.2 Attack on Time Series Forecasting

Adversarial attacks in time series forecasting have emerged as a critical research focus, exposing the vulnerabilities of forecasting models. Unlike static domains such as image recognition, time series forecasting presents unique challenges for adversarial research. One key constraint is the inability to use future ground truth values when generating perturbations, as this could lead to information leakage (Liu et al., 2022a). To address this, surrogate modeling techniques have been introduced (Liu et al., 2021), enabling attackers to bypass the need for ground truth labels.

Most prior studies have concentrated on white-box scenarios, where adversaries have full access to model parameters. These investigations have demonstrated that even small input disruptions can cause significant drops in forecasting accuracy (Liu et al., 2023). However, evaluating the robustness of LLM-based forecasting models presents additional complexities. These models typically operate in black-box settings, limiting access to their internal workings. Gradient-free black-box attacks have been proposed as a solution (Liu et al., 2024b), but they often require modifying the entire time series, which is impractical for real-world applications.

## 3 LLM-Based Time Series Forecasting

LLMs have shown great promise in time series forecasting by leveraging their next-token prediction capability. A typical LLM-based time series forecasting framework, denoted as $f(\cdot)$, comprises two key components: an embedding or tokenization module and a pre-trained LLM. The embedding module encodes time series into a sequence of tokens suitable for processing by the LLM, while the LLM captures temporal dependencies and autoregressively predicts subsequent tokens based on its learned representations.

Let $\mathbf{X}_t \in \mathbb{R}^d$ represent a $d$-dimensional time series at time $t$. Define $\mathcal{X}_t = \{\mathbf{X}_{t-T+1}, \ldots, \mathbf{X}_t\}$ as the sequence of $T$ recent historical observations and $\mathcal{Y}_t = \{\mathbf{Y}_{t+1}, \ldots, \mathbf{Y}_{t+L}\}$ as the true future values for the next $L$ time steps. The forecasting model $f(\cdot)$ predicts the future values from the historical observations, which is formulated as:

$$\hat{\mathcal{Y}}_t = f(\mathcal{X}_t), \tag{1}$$

where $\hat{\mathcal{Y}}_t$ denotes the predicted future values. Typically, the prediction horizon $L$ is constrained to be less than or equal to the historical horizon $T$, i.e., $L \leq T$. This ensures that the model leverages sufficient historical context while maintaining computational efficiency.

By effectively combining the embedding module's ability to encode raw time series data and the LLM's capacity to model complex temporal patterns, these models have become powerful tools for addressing a wide range of forecasting challenges across various domains.

## 4 Threat Model

The goal of attacking an LLM-based time series forecasting model $f(\cdot)$ is to manipulate it into producing abnormal outputs that differ substantially from their typical predictions and the actual ground truth, using minimal and nearly undetectable perturbations.

The adversarial attack can be modeled as a maximum optimization problem:

$$\max_{\boldsymbol{\rho}} \mathcal{L}\left(f\left(\mathcal{X}_t + \boldsymbol{\rho}\right), \mathcal{Y}_t\right)$$
$$\text{s.t.} \quad \|\rho_i\|_p \leq \epsilon, i \in [t - T + 1, t],$$

(2)

where $\boldsymbol{\rho} = \{\rho_{t-T+1}, \ldots, \rho_t\}$ denotes the perturbations added into the clean historical time series $\mathcal{X}_t = \{\mathbf{X}_{t-T+1}, \ldots, \mathbf{X}_t\}$, and $\mathcal{Y}_t = \{\mathbf{Y}_t, \ldots, \mathbf{Y}_{t+L}\}$ represents the true future values of the subsequent $L$ time steps. Here, the loss function $\mathcal{L}$ measures the discrepancy between the model's predictions and the ground truth, while $\epsilon$ serves as a constraint on the perturbation magnitude under the $\ell_p$-norm, ensuring that the adversarial attack remains subtle and imperceptible. Typically, the global average $\bar{\mathcal{X}}$ serves as the reference point to determine whether the added perturbations are imperceptible. Consequently, $\epsilon$ is defined as a proportion of the global average, e.g., $\epsilon = 5\% \times \bar{\mathcal{X}}$.

The true future values $\mathcal{Y}_t$ are generally unavailable during the practical forecasting process. For example, in a 5-minute-ahead Google stock value prediction, the ground truth of the stock value at 10:00 am corresponds to its value at 10:05 am, which remains inaccessible to both the forecaster and the attacker. As a result, to avoid future information leakage, the ground truth $\mathcal{Y}_t$ is substituted with the predicted values $\hat{\mathcal{Y}}_t$ produced by the forecasting model. Specifically, in Eq. equation 2, $\mathcal{Y}_t$ is replaced with $\hat{\mathcal{Y}}_t$. In practical applications, it is generally infeasible to access the complete set of detailed parameters of an LLM, compelling the attacker to approach the target model as a black-box system. In other words, no internal information of $f(\cdot)$ in Eq. equation 2 is available.

The computed perturbations $\boldsymbol{\rho} = \{\rho_{t-T+1}, \ldots, \rho_t\}$ are typically applied across the entire time series, making the poisoning process highly challenging for attackers. In this study, we impose strict limitations on the attacker's capabilities, allowing them to pollute only $\tau$ time steps within the input time series. Furthermore, since the future true values $\mathcal{Y}_t$ are unavailable, they are approximated using the predicted values $\hat{\mathcal{Y}}_t = f(\mathcal{X}_t)$. Under this constraint, the attack process is reformulated as a CCOP (Bhattacharya, 2009):

$$\max_{\boldsymbol{w}} \mathcal{L}\left(f\left(\mathcal{X}_t\left(1 + \boldsymbol{w}\right)\right), \hat{\mathcal{Y}}_t\right)$$
$$\text{s.t.} \quad \|\boldsymbol{w}\|_0 = \tau,$$
$$\|w_i\|_1 \leq \epsilon, \quad i \in [t - T + 1, t],$$

(3)

where $\boldsymbol{w} = \{w_{t-T+1}, \ldots, w_t\}$ represents multiplicative adversarial perturbations. The cardinality constraint, also called $\tau$-sparse $\ell_0$-norm constraint, restricts the number of non-zero elements in adversarial perturbations to a fixed small number, ensuring that the adversarial perturbations are sparse on the temporal dimension. Besides, the $\ell_1$-norm constraint limits the magnitude of each non-zero perturbation, ensuring the modifications remain imperceptible.

It should be noted that the global average is unsuitable as a reference for the average magnitude of the manipulated series under the temporally sparse setting. Instead, each manipulated time step requires a unique reference point to ensure the magnitude of the perturbation at each time step is bounded. The limitation of the poisoned value at time step $i$ can be expressed as:

$$\|\mathbf{X}_i + \rho_i\|_1 = \|\mathbf{X}_i\left(1 + w_i\right)\|_1 \leq \|\mathbf{X}_i\left(1 + \epsilon\right)\|_1,$$

(4)

where $\|\rho_i\|_1 = \|w_i \cdot \mathbf{X}_i\|_1 \leq \|\epsilon \cdot \mathbf{X}_i\|_1$. Consequently, the additive perturbation $\mathcal{X}_t + \boldsymbol{\rho}$ in Eq. equation 2 is replaced with the multiplicative perturbation $\mathcal{X}_t\left(1 + \boldsymbol{w}\right)$ in Eq. equation 3.

Additionally, in many real-world scenarios, attackers lack access to the complete training dataset, making it impractical for them to exploit training data directly. Based on previous discussion, the attacker's capabilities and limitations in this context can be summarized as follows:

- No access to the training data;
- No access to the internal structure or parameters of the LLM-based forecasting model;
- No access to the ground truth values;
- No ability to manipulate the entire time series data;
- Limited to temporally sparse manipulations;
- Possesses the ability to query the target model.

# 5 PERTURBATION COMPUTATION WITH SUBSPACE PURSUIT

## 5.1 SINGLE-STEP PERTURBATION WITH ZERO OPTIMIZATION

Before solving the optimization problem in Eq. equation 3 to generate $\tau$-sparse perturbations, we first consider generating a perturbation at the specific time step $i$. This can be formulated as:

$$\max_{w_i} \mathcal{L}\left(f\left(\mathcal{X}_t + \{0, \ldots, w_i \cdot \mathbf{X}_i, \ldots, 0\}\right), \hat{\mathcal{Y}}_t\right)$$
$$\text{s.t. } \|w_i\|_1 \leq \epsilon. \tag{5}$$

Here, the perturbation $w_i$ is applied only at time step $i$. The magnitude of the perturbation is bounded by the constraint $\epsilon$, while maximizing the impact on the loss function $\mathcal{L}$.

In the black-box setting, Eq. equation 5 cannot be solved using gradient-based methods such as Stochastic Gradient Descent (SGD). Instead, a zero optimization technique can be employed to estimate the gradients, as follows:

$$\hat{g} = \frac{\mathcal{F}(\mathcal{X}_t, w_i, \Delta) - \mathcal{F}(\mathcal{X}_t, w_i, -\Delta)}{2 \cdot \Delta}, \tag{6}$$

where $\hat{g}$ represents the estimated gradients, $\Delta$ denotes a random Gaussian noise, and $\mathcal{F}(\mathcal{X}_t, w_i, a) = f\left(\mathcal{X}_t + \{0, \ldots, (w_i + a) \cdot \mathbf{X}_i, \ldots, 0\}\right)$ denotes querying the target forecasting model with a noise term $a$.

Similar to the Fast Gradient Sign Method (FGSM) (Goodfellow et al., 2015), the perturbation can be computed using the estimated gradients $\hat{g}$ as follows:

$$w_i = \epsilon \cdot \text{sign}\left(\hat{g}\right), \tag{7}$$

where $\text{sign}(\cdot)$ denotes the signum function. This approach ensures that the perturbation magnitude is bounded by $\epsilon$ while aligning with the direction of the estimated gradients.

Combining Eq. equation 6 and Eq. equation 7 offers an effective approach for computing single-step perturbations in a black-box setting, where direct access to the model's internal parameters is restricted. However, Eq. equation 3 (a CCOP) is still not solved as it cannot strictly limit the number of non-zero elements in the perturbations. To overcome this limitation, we propose an SP-based algorithm (detailed in Algorithm 1) where the zero optimization-based method is embedded as a submodule.

## 5.2 $\tau$-SPARSE PERTURBATION COMPUTATION

To solve the optimization problem in Eq. equation 3, it is essential to ensure both the sparsity of the perturbation vector $\boldsymbol{w}$ and the bounded magnitude of its elements. In this study, we propose an adapted SP method, as outlined in Algorithm 1, based on the approach by dai2009subspace. In our adaption, the $\ell_1$-norm constraint is incorporated as a subroutine to maintain the imperceptibility of the perturbations. Here, the support set $S = \text{supp}(\boldsymbol{w}) = \{i : w_i \neq 0\}$ denotes the indices of nonzero elements in the perturbation vector $\boldsymbol{w}$, with $|S|$ representing its cardinality. To efficiently update the support set, we define the merge operator:

$$\mathcal{M}\left(\boldsymbol{w}_S, w_j\right) = \begin{cases} \boldsymbol{w}_S, & j \in S, \\ \{\boldsymbol{w}_S, w_j\}, & j \notin S. \end{cases} \tag{8}$$

This operator ensures that when a new candidate perturbation $w_j$ is selected, it is either retained in the existing support set $S$ if it is already present or added as a new element if it is not.

Algorithm 1 describes the iterative process for estimating the sparse multiplicative adversarial perturbations $\boldsymbol{w}$. At each iteration, the algorithm identifies the indices corresponding to the $\tau$ largest loss values resulting from applying candidate perturbations. The individual perturbations $w_j$ are computed using the zero optimization technique in Eq.equation 6 and Eq.equation 7. Then, the support set $S$ is updated by including the identified indices. The support set $S$ is subsequently refined by selecting the $\tau$ elements with the largest individual prediction loss. Any perturbation components outside the updated support set are reset to zero. This process repeats until the loss $\boldsymbol{r}$ converges and the final $\tau$-sparse multiplicative adversarial perturbation $\boldsymbol{w}$ is returned.

---

**Algorithm 1** Computing $\boldsymbol{w}$ with adapted SP

---

1: **Input:** Time series $\mathcal{X} \in \mathbb{R}^{d \times T}$, the loss function $\mathcal{L}$, the LLM-based forecaster $f(\cdot)$, and sparsity level $\tau$ of the multiplicative adversarial perturbations $\boldsymbol{w}$.
2: **Initialize** the perturbation vector $\boldsymbol{w} := \boldsymbol{0}$ as zeros, the support set $S := \emptyset$ as an empty set, and the loss value $\boldsymbol{r} := 0$ as zero.
3: **while** not converged **do**
4:     Find $\ell$ as the index set of the $\tau$ largest losses of $f\left(\mathcal{X}_t\left(1 + \mathcal{M}\left(\boldsymbol{w}_S, w_j\right)\right)\right)$ in which $w_j$ is computed separately following Eq. equation 6 and Eq. equation 7, where $j \in [1, \ldots, T] \;\&\; j \notin S$.
5:     Update the support set $S := S \cup \{\ell\}$.
6:     Update the sparse vector $\boldsymbol{w}_S := \epsilon \cdot \text{sign}\left(\hat{\boldsymbol{g}}_S\right)$.
7:     Update the support set $S$ as the index set of the $\tau$ largest losses of $f\left(\mathcal{X}_t\left(1 + w_i\right)\right)$ for all $i \in S$.
8:     Set $w_i = 0$ for all $i \notin S$.
9:     Update $\boldsymbol{r} := \mathcal{L}\left(f\left(\mathcal{X}_t\left(1 + \boldsymbol{w}_S\right)\right), \hat{\mathcal{Y}}_t\right)$.
10: **end while**
11: Return the $\tau$-sparse multiplicative adversarial perturbations $\boldsymbol{w}$.

---

This method effectively enforces the CCOP by ensuring that only $\tau$ time steps are modified while maintaining a bounded perturbation magnitude. The adapted SP approach enables efficient selection of perturbation locations, ensuring maximal adversarial impact while keeping modifications imperceptible. Moreover, the computation complexity of the proposed method is $\mathcal{O}\left(T \times \tau\right)$, whereas a standard greedy algorithm has a significantly higher complexity of $\mathcal{O}\left(T^\tau\right)$.

# 6 EXPERIMENT

## 6.1 DATASETS

To assess the effectiveness of the temporally sparse attack and evaluate the robustness of LLM-based forecasting models, we utilized four real-world time series datasets:

- **ETTh1** (Zhou et al., 2021): Hourly temperature and power consumption data from electricity transformers recorded over two years, capturing both seasonal trends and long-term variations.
- **IstanbulTraffic** (Gruver et al., 2024): Hourly traffic volume data from Istanbul, reflecting dynamic temporal dependencies influenced by traffic flow fluctuations and congestion cycles.
- **Weather** (Zhou et al., 2021): Hourly meteorological data, including temperature, humidity, and wind speed, which poses forecasting challenges due to high variability and nonlinear patterns.
- **Exchange Rates** (Lai et al., 2018): Daily foreign exchange rate data for eight countries from 1990 to 2016, providing insights into long-term economic trends and temporal dependencies.

For all datasets, the data was split into 60% for training, 20% for validation, and 20% for testing. The adversarial attacker had no access to the training or validation data, ensuring a realistic black-box setting. All forecasting models were trained using a 96-step historical input window to predict the next 48 steps, maintaining consistency across experiments.

## 6.2 TARGET MODELS

Three representative LLM-based forecasting models, along with one transformer-based forecasting model, are included in the experiment to assess the effectiveness of TSA:

- **TimeGPT** (Garza & Mergenthaler-Canseco, 2023): A pre-trained LLM specialized for time series forecasting, incorporating advanced attention mechanisms and temporal encoding to capture complex patterns.
- **LLMTime** (Gruver et al., 2024): A general-purpose LLM adapted for time series forecasting by framing it as a next-token prediction task. We evaluate multiple versions, including those based on GPT-3.5, GPT-4, LLaMA, and Mistral.

Table 1: Univariate time series forecasting results with an input length of 96 and output length of 48. Lower MSE and MAE indicate better performance. The sparsity level $\tau$ is set to 9, TSA magnitude constraint $\epsilon$ to 0.1, and GWN deviation to 2% of each dataset's mean. Bold values denote the worst performance for each dataset-model combination.

| Models | LLMTime w/ GPT-3.5 | | LLMTime w/ GPT-4 | | LLMTime w/ LLaMa 2 | | LLMTime w/ Mistral | | TimeLLM w/ GPT-2 | | TimeGPT (2024) | | TimesNet (2023) | |
|---|---|---|---|---|---|---|---|---|---|---|---|---|---|---|
| Metrcis | MSE | MAE | MSE | MAE | MSE | MAE | MSE | MAE | MSE | MAE | MSE | MAE | MSE | MAE |
| ETTh1 | 0.073 | 0.213 | 0.071 | 0.202 | 0.086 | 0.244 | 0.097 | 0.274 | 0.089 | 0.202 | 0.059 | 0.192 | 0.073 | 0.202 |
| w/ GWN | 0.077 | 0.219 | 0.076 | 0.213 | 0.087 | 0.237 | 0.094 | 0.291 | **0.102** | 0.231 | 0.059 | 0.193 | 0.074 | 0.202 |
| w/ TSA | **0.082** | **0.235** | **0.079** | **0.230** | **0.092** | **0.249** | **0.097** | **0.295** | 0.091 | **0.237** | **0.061** | **0.203** | **0.080** | **0.206** |
| IstanbulTraffic | 0.837 | 0.844 | 0.805 | 0.779 | 0.891 | 1.005 | 0.826 | 0.973 | 0.995 | 1.013 | 1.890 | 1.201 | 1.095 | 1.022 |
| w/ GWN | 0.882 | 0.908 | 0.883 | 0.864 | 0.917 | 1.063 | 1.054 | 1.031 | 1.123 | 1.221 | 1.848 | 1.204 | 1.103 | 1.035 |
| w/ TSA | **0.901** | **1.037** | **1.179** | **1.008** | **0.969** | **1.085** | **1.493** | **1.204** | **1.147** | **1.332** | **1.920** | **1.208** | **1.136** | **1.093** |
| Weather | 0.005 | 0.051 | 0.004 | 0.048 | 0.008 | 0.072 | 0.006 | 0.057 | 0.004 | 0.034 | 0.004 | 0.043 | 0.003 | 0.042 |
| w/ GWN | 0.005 | 0.053 | 0.005 | 0.051 | 0.008 | 0.074 | **0.007** | **0.066** | 0.004 | 0.033 | 0.004 | 0.043 | 0.003 | 0.042 |
| w/ TSA | 0.005 | **0.060** | **0.006** | **0.058** | **0.010** | **0.076** | 0.006 | 0.065 | **0.004** | **0.048** | **0.007** | **0.072** | **0.004** | **0.043** |
| Exchange | 0.038 | 0.146 | 0.040 | 0.152 | 0.043 | 0.167 | 0.151 | 0.274 | 0.056 | 0.188 | 0.256 | 0.368 | 0.056 | 0.184 |
| w/ GWN | 0.042 | 0.179 | 0.046 | 0.182 | 0.050 | 0.185 | 0.160 | 0.298 | 0.059 | **0.194** | 0.329 | 0.413 | **0.065** | **0.195** |
| w/ TSA | **0.049** | **0.196** | **0.065** | **0.190** | **0.059** | **0.210** | **0.190** | **0.299** | **0.061** | 0.189 | **0.474** | **0.537** | 0.062 | 0.190 |

- **TimeLLM** (Jin et al., 2024): A model that reprograms time series data into textual inputs for LLMs, leveraging the Prompt-as-Prefix (PaP) technique to enhance forecasting accuracy.
- **TimesNet** (Wu et al., 2023): A non-LLM transformer-based forecasting model introduced to explore the potential impact of our attack on non-LLM models.

These models represent three key strategies for time series forecasting: (1) domain-specific pre-training tailored for time series data (TimeGPT), (2) adapting general-purpose LLMs to forecasting tasks (LLMTime), and (3) input reprogramming to enhance compatibility with LLMs (TimeLLM). Additionally, the inclusion of a non-LLM model (TimesNet) provides a broader framework for evaluating adversarial robustness across both LLM-based and non-LLM models.

## 6.3 SETUP

We conducted experiments to evaluate TSA's effectiveness on LLM-based forecasting models using various datasets. The process involved (i) applying TSA to mislead forecasts while preserving time series structure, (ii) introducing Gaussian White Noise (GWN) as a baseline, and (iii) measuring performance degradation with Mean Absolute Error (MAE) and Mean Squared Error (MSE). Experiments were performed on Ubuntu 18.04 LTS with PyTorch 1.7.1, Python 3.7.4, and a Tesla V100 GPU.

## 6.4 OVERALL COMPARISON

As shown in Table 1, TSA significantly increases both MSE and MAE across models and datasets, demonstrating its strong adversarial impact. TSA causes greater prediction errors than GWN, with the IstanbulTraffic dataset showing the largest increase—80.75% for LLMTime w/ Mistral and 46.45% for LLMTime w/ GPT-4—highlighting model vulnerability.

Figure 2 compares prediction errors under TSA and GWN for LLMTime w/ GPT-3.5 and TimeGPT. Subfigures 2(a) and 2(c) show that TSA-induced deviations from the ground truth are larger than those caused by GWN. Subfigures 2(b) and 2(d) reveal that TSA (orange) generates significantly higher error regions than GWN (purple), confirming TSA's stronger adversarial effect.

These results validate TSA's effectiveness. Manipulating only 9 out of 96 time steps, TSA outperforms GWN, which affects all steps, demonstrating the power of sparse perturbations. Hyperparameter analysis for $\tau$ and $\epsilon$ is detailed in Section 6.7.

## 6.5 INTERPRETATION

Figure 3 illustrates the impact of TSA on LLMTime with GPT-3.5 using the ETTh1 dataset. Subfigures 3(a) and 3(b) compare input and output distributions under clean input (orange), GWN (blue), and TSA (pink). While the input distributions show minor differences across all cases, the output

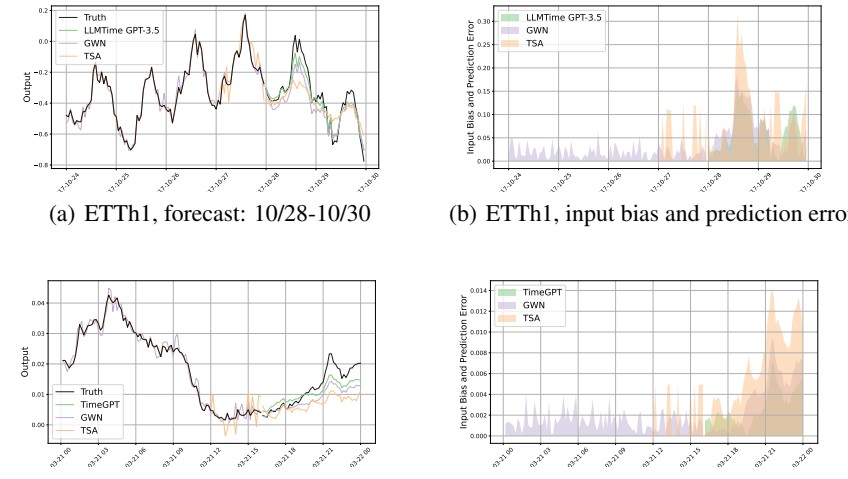

(a) ETTh1, forecast: 10/28-10/30  (b) ETTh1, input bias and prediction error

(c) Weather, forecast: 3/21/16pm,-3/22/12am  (d) Weather, input bias and prediction error

Figure 2: Comparison of prediction errors and input bias for LLM-Time with GPT-3.5 and TimeGPT under TSA and GWN. This figure illustrates the greater impact of TSA, demonstrating significant deviations from the ground truth compared to GWN.

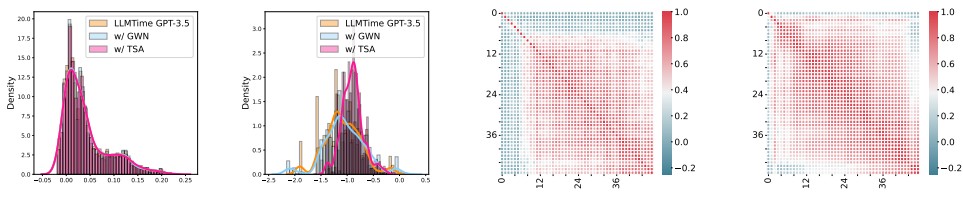

(a) Input Distribution  (b) Output Distribution (c) Clean Error Correlation (d) Poisoned Error Correlation

Figure 3: (a) and (b) compare the input and output distributions for LLMTime with GPT-3.5 on ETTh1 under clean input (orange), GWN (blue), and the proposed TSA (pink). While the input distributions remain relatively similar across all cases, the output distribution under TSA deviates more significantly compared to those under clean input and GWN. (c) and (d) show the correlation matrices of prediction errors with and without the proposed attack.

distribution under TSA deviates significantly, indicating that TSA exerts a stronger adversarial effect than GWN by disrupting model forecasts more severely.

Subfigures 3(c) and 3(d) show the correlation matrices of prediction errors for clean and attacked scenarios. The matrix under attack 3(d) exhibits higher error correlations, suggesting that TSA induces structured perturbations that propagate across the forecast horizon. This highlights that TSA causes systematic distortions rather than random noise, leading to more pronounced forecasting errors.

## 6.6 ATTACK DEFENDED LLM-BASED FORECASTING MODELS

This section evaluates TSA's effectiveness against adversarial defenses in LLM-based forecasting. A gradient-free full-series attack (Liu et al., 2024b), with perturbations scaled to 2% of the dataset mean, serves as a baseline. Three filter-based defenses—Gaussian, Mean, and Quantile filters (Xie et al., 2019)—are applied without model re-training.

Figure 4 shows that these defenses fail to mitigate TSA (minimal light orange bars) but effectively reduce errors for full-series attacks (larger light green bars). TSA's sparse, targeted perturbations

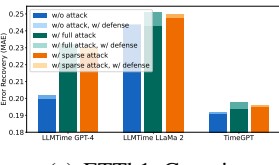 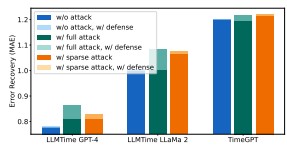 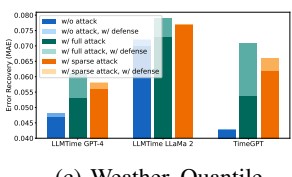

(a) ETTh1, Gaussian  (b) IstanbulTraffic, Mean  (c) Weather, Quantile

Figure 4: Full series and temporally sparse adversarial attacks on different LLM-based forecasting models protected by filter-based adversarial defense strategies. Light green and light orange indicate the recovered error.

bypass statistical assumptions underlying these defenses, introducing structured errors that persist across the forecast horizon and significantly degrade model performance.

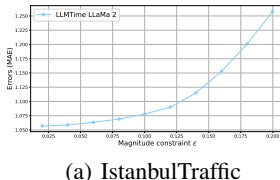 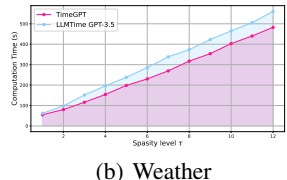 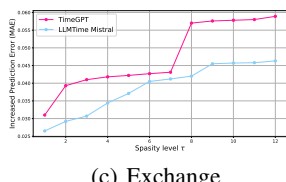

(a) IstanbulTraffic  (b) Weather  (c) Exchange

Figure 5: Hyperparameter analysis. (a) Prediction errors for LLMTime with LLaMa 2 on Istanbul-Traffic grow exponentially with $\epsilon$. (b) Computational cost scales linearly with $\tau$. (c) Prediction errors for TimeGPT and LLMTime with Mistral increase with higher $\tau$.

### 6.7 HYPERPARAMETER ANALYSIS

Algorithm 1 has two key hyperparameters: perturbation magnitude $\epsilon$ and sparsity level $\tau$. Figure 5 illustrates their impact on TSA's effectiveness and computational cost.

Subfigure 5(a) shows that prediction errors for LLMTime with LLaMa 2 on IstanbulTraffic increase exponentially with higher $\epsilon$, reflecting a trade-off between attack effectiveness and imperceptibility. Subfigure 5(b) indicates that TSA's computational cost scales linearly with $\tau$, as more perturbed steps lead to proportional increases in processing time. Subfigure 5(c) highlights that prediction errors for TimeGPT and LLMTime with Mistral rise with higher $\tau$, with TimeGPT showing a stronger error increase. These results reveal a trade-off between attack impact and computational complexity.

## 7 CONCLUSION

This work presents a Temporally Sparse Attack (TSA), designed for LLM-based time series forecasting models in constrained adversarial scenarios, where only a small subset of input time steps can be modified. We model the attack as a Cardinality-Constrained Optimization Problem (CCOP) and develop a Subspace Pursuit (SP)-based method to efficiently generate sparse perturbations. Our approach operates in a black-box setting, requiring no access to future data or internal model parameters.

Experiments on three advanced LLM-based time series forecasting models across diverse real-world datasets show that perturbing only a small portion of input time steps significantly degrades forecasting performance. Both large pre-trained models and fine-tuned models exhibit high sensitivity to adversarial manipulation. Our findings also demonstrate that conventional filter-based approaches fail to mitigate TSA, emphasizing the importance of enhancing robustness in time series foundation models. This research provides a framework for improving the resilience of AI systems and supports future advancements in Trustworthy AI.

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
