# OpenReview forum: "Temporally Sparse Attack for Fooling Large Language Models in Time Series Forecasting"
_ICLR.cc/2025/Workshop/BuildingTrust — BuildingTrust_

### Official Review · Reviewer_uycS · 2025-02-25
**Temporally Sparse Attack for Fooling Large Language Models in Time Series Forecasting**

**Rating:** 7
**Confidence:** 3

**Review:**

**Pros**:

* The experimental evaluation is comprehensive.
* The logics and explanation is clear to me.

**Cons**:
* From my perspective, the proposed attack is basically a gradient-based optimization with sparsity constraints. The novelty is not enough
* The writing need to improve. For example, in experimental part, the content is not well-organized due to too many subsections.
* The font size of the figures is not large enough.

---

### Official Review · Reviewer_UpiJ · 2025-02-28
**This paper presents a temporally sparse adversarial attack against LLM-based time series forecasting, revealing critical vulnerabilities but lacking discussion on attack transferability and defense strategies.**

**Rating:** 8
**Confidence:** 5

**Review:**

This work introduces Temporally Sparse Attack (TSA), an adversarial method targeting LLM-based time series forecasting models by modifying only a small fraction of input data. Its very clear the topic here is very relevant to this workshop as well as important considering the current industry scenario. The attack is formulated as a Cardinality-Constrained Optimization Problem (CCOP) and solved using Subspace Pursuit (SP), ensuring minimal yet high-impact perturbations, This looks promising. The paper is well-structured with strong theoretical grounding, rigorous empirical validation (Sign of a good paper) , and a clear black-box threat model which i really liked and appreciate that authors didn't left it out. The results convincingly show that perturbing just 10% of the input can significantly degrade performance on LLMTime (GPT-3.5, GPT-4, LLaMa, Mistral), TimeGPT, and TimeLLM, highlighting a fundamental weakness in LLM-based forecasting. However, the study does not explore attack transferability to unseen models, evaluate computational efficiency, or propose meaningful defense mechanisms. A deeper analysis of these aspects would significantly improve the impact of this work.

Strengths
1. Novel sparse adversarial attack with strong empirical validation
2. Rigorous black-box formulation using CCOP and SP
3. Reveals critical security risks in LLM-based forecasting
4. Well-structured experimental design

Weaknesses
1. Lack of discussion on attack transferability
2. Computational efficiency of SP-based attack remains unclear
3. Limited exploration of defense strategies

---

### Decision · Program_Chairs · 2025-03-04

Accept